# Combination of Anti-Cancer Drugs with Molecular Chaperone Inhibitors

**DOI:** 10.3390/ijms20215284

**Published:** 2019-10-24

**Authors:** Maxim Shevtsov, Gabriele Multhoff, Elena Mikhaylova, Atsushi Shibata, Irina Guzhova, Boris Margulis

**Affiliations:** 1Center for Translational Cancer Research, TUM (TranslaTUM), Technische Universität München (TUM), Klinikum rechts der Isar, Radiation Immuno Oncology, Einsteinstr. 25, 81675 Munich, Germany; gabriele.multhoff@tum.de; 2Institute of Cytology of the Russian Academy of Sciences (RAS), Tikhoretsky Ave., 4, St. Petersburg 194064, Russia; mikhailovaer@yandex.ru (E.M.); irina.guzh@gmail.com (I.G.); 3Department of Biotechnology Pavlov First Saint Petersburg State Medical University, L’va Tolstogo str., 6/8, St. Petersburg 197022, Russia; 4Almazov National Medical Research Centre, Polenov Russian Scientific Research Institute of Neurosurgery, Mayakovskogo str., 12, St. Petersburg 191014, Russia; 5Department of Biomedical Cell Technologies Far Eastern Federal University, Russky Island, Vladivostok 690000, Russia; 6National Center for Neurosurgery, Turan Ave., 34/1, Nur-Sultan 010000, Kazakhstan; 7Signal Transduction Program, Gunma University Initiative for Advanced Research (GIAR), Gunma University, Maebashi, Gunma 371-8511, Japan; shibata.at@gunma-u.ac.jp

**Keywords:** heat shock protein inhibitors, molecular chaperones, HSP90, HSP70, HSP27, concurrent therapy, cancer therapy

## Abstract

Most molecular chaperones belonging to heat shock protein (HSP) families are known to protect cancer cells from pathologic, environmental and pharmacological stress factors and thereby can hamper anti-cancer therapies. In this review, we present data on inhibitors of the heat shock response (particularly mediated by the chaperones HSP90, HSP70, and HSP27) either as a single treatment or in combination with currently available anti-cancer therapeutic approaches. An overview of the current literature reveals that the co-administration of chaperone inhibitors and targeting drugs results in proteotoxic stress and violates the tumor cell physiology. An optimal drug combination should simultaneously target cytoprotective mechanisms and trigger the imbalance of the tumor cell physiology.

## 1. Overview on Combinational Anti-Cancer Therapies

In the last decade, progress in the development of anti-tumor therapies has been achieved, especially with respect to specifically targeting anti-tumor drugs that aim to interfere with molecular mechanisms that are responsible for tumor growth. However, traditional chemotherapeutic agents such as cisplatin, etoposide, 5-fluorouracil and their derivatives as well as ionizing radiation, are still widely applied to cure cancer. Both strategies can exert harmful effects on normal cells and may cause serious changes in the tumor microenvironment, which makes it difficult to predict outcome of these approaches.

To improve tumor control, combinatorial therapies that target a wide range of tumorigenic functions including extensive cell growth, motility, proteostasis, apoptosis, autophagy and others are presently under investigation (see Figure 1). The concurrent administration of several drugs that are targeting different cellular drivers of tumor cell viability and propagation is often advantageous over monotherapeutic approaches because they might act synergistically. Furthermore, in combinatorial therapies, reduced doses of the individual drugs can be applied and therefore, unfavorable side effects can be avoided [1,2]. Currently, bioinformatic models have been designed that might predict synergistic effects of novel drug combinations [3,4,5]. Combinatorial formulations could also exert beneficial effects in therapy-resistant tumor cells by down-regulating different metabolic factors, which are induced by i.e., hypoxia, oxidative or reductive stress. Thirdly, combinatorial therapies may reduce tumor recurrence by suppression of survival mechanisms in resistant tumor cells after first-line therapy [6]. The relevance of combinational therapies in oncology is documented by a large number of clinical trials which are registered at clinicaltrials.gov and by 2548 references which appear when the two terms “combination therapy” and “cancer, metastasis” are entered.

Drugs target polypeptides, protein modifications like phosphorylation, proteolysis or acetylation, nucleic acids (i.e., RNA and DNA), lipids, and cancer metabolites and thereby impact on mitochondria, membranes, nucleus, ER and many other subcellular targets (Figure 1). By damaging multiple cellular functions via combinatorial therapies alternative cell death mechanisms such as senescence or autophagy can be induced or occasionally a conversion to a more resistant, stem-like phenotype can occur [7,8]. The search for a successful combinatorial therapy may start with a better understanding of molecular mechanisms of a drug hitting a certain target. Connectivity Map is a platform designed to provide information on signaling pathways induced by a certain drug [9]. Another powerful tool revealing clinically tested drug combinations is SynTarget which is based on 15 large-scale gene expression data sets covering eight different cancer types.

The list of components of therapeutic combinations includes chemicals (small molecules), proteins, particularly antibodies, physico-chemical interventions like radiation or hyperthermia, photodynamic treatment, anti-cancer vaccines and some other agents (Figure 1; Table 1). Since therapies should specifically target tumor cells and not the surrounding normal tissue, special constructs are designed for a targeted delivery of the drug. Some of those vehicles are based on nanoparticles composed of a magnetic or silicone core and specific peptides (aptamers) or antibodies that are directed against cell surface molecules on cancer cells. A brief overview on anti-cancer therapies indicates that a specific spectrum of drugs is necessary to cure certain types of cancer. For instance, for the optimal treatment of prostate cancer cells a set of different compounds such as mTOR-inhibitor RAD001, the dual tyrosine kinase inhibitor AEE788 and the histone deacetylase inhibitor, valproic acid should be applied. The cocktail was found to be more efficient in reducing tumor cell growth, adhesion and migration than each of the drugs alone [10].

A long list of tyrosine kinases inhibitors and their combinations is already used in oncology, and many more are tested in preclinical studies. Examples for such applications are AC220 Quizartinib FLT3 tyrosine kinase and TAK-165 kinase inhibitors applied as small molecules [11], as shown in Table 1. In another study two modulators of completely distinct signaling pathways, PAC-1 pro-caspase-3 activator and vemurafenib, inhibitor of BRAF oncogene, whose mutation is a common cause of melanoma, demonstrated strong therapeutic activity in prevention of resistance to MAPK inhibitors and tumor regrowth in preclinical melanoma models [12]. Combinations of anti-cancer drugs are often packaged in special containers to preserve them from degradation or chemical modifications. An example for such a construct is based on nanoparticles consisting of α-tocopherol and polyethylene glycol linked to poly(L-glutamic acid). Docetaxel and cisplatin are often loaded into micelles which were coated with an αvβ3 integrin targeting peptide to specifically address cancer cells. These dual drug-loaded micelles showed anti-tumor and anti-metastatic efficacy and had a remarkable long circulation time [14] (Table 1).

Small molecule inhibitors are often combined with monoclonal antibodies such as erlotinib and the monoclonal antibody Erb/HER which neutralize tyrosine kinases and demonstrated efficacy in non-small cell lung cancer (NSCLC) in vitro and in vivo [15]. Another example is the combination of an antibody directed against mutated KRAS oncogene, a common cause of pancreatic cancer with gemcitabine. This combination inhibited angiogenesis, migration, and invasion of tumor cells and showed significant anti-cancer activity [16]. Many successful new combinations use monoclonal antibodies inhibiting immune checkpoints, such as CTLA-4 or PD-1 receptors (i.e., trametinib, dabrafenib) and MEK and BRAF inhibitors, respectively [18]. These combined therapies show enhanced anti-metastatic effects and an enhanced anti-tumor immunity in clinical trials [22].

Besides drugs oxidative stress was also used to damage tumor cells. As an example, oxidative stress induced by physical exercises in combination with temozolomide showed a reduction in metastases in a few patients with glioblastoma [20]. Oncolytic viruses and paclitaxel encapsulated in extracellular vesicles resulted in enhanced anti-tumor effects in an animal model of lung cancer [21]. The progress of oncolytic virus-mediated therapies combined with anti-cancer drugs and the selection of effective protocols is discussed in a recent publication [21]. Lastly, despite obvious problems in the clinical application of gene therapies in clinical practice, a recent review reported on the application of a siRNA approach in combination with traditional anti-cancer drugs [23].

In conclusion, the list of therapeutic tools applied in combinations includes standard drugs, specific antibodies, oncolytic viruses, si- or sh-RNA, ultrasound radiation, local hyperthermia, exercise and others. There are technologies available to packaging drugs, virus particles or antibodies into multifunctional nanoparticles, liposomes or exosomes, and these constructs can be targeted specifically to cancer cells via tumor-targeting peptides or antibodies. We found 133 records at the site clinicaltrials.gov using the key words “nano-particle” and “cancer”. In a stage II clinical trial (clinicaltrials.gov identifier NCT03531827) a formulation was used which consisted of Enzalutamide serving as a first-line anti-hormonal therapy for prostate cancer patients combined with CRLX101 nanoparticle drug conjugates composed of camptothecin, a highly selective topoisomerase I inhibitor with anti-HIF-1α properties. Another complex which is also tested in a phase II trial to target pancreatic duct carcinoma is composed of three well known anti-cancer drugs, nanoparticle albumin-bound paclitaxel, gemcitabine, and cisplatin (Clinical.Trials.gov identifier NCT03410030). Abraxane (albumin-bound paclitaxel) combined with gemcitabine showed a prolonged overall survival and a delayed tumor growth in patients with pancreatic adenocarcinoma.

## 2. Molecular Chaperones as Drug Targets in Combinational Therapy

### 2.1. Inhibitors of HSF in Anti-Cancer Therapy Formulations

The heat shock response is a thoroughly regulated mechanism based on the activation of different heat shock transcription factors [24,25]. The activity of HSF1 is controlled by the phosphorylation of certain serine and threonine residues, acetylation of lysines, trimerization and its transport into the nucleus [26]. Activated, trimeric HSF1 recognizes the heat shock element (HSE) in the promoter region of genes coding for HSPs and regulates their transcription. Based on their molecular weights, HSPs can be divided into different families: small HSPs (HSP27, crystallins), J-domain proteins (DNAJ, HSP40), HSP60, HSP70, HSP90 and HSP110 (Figure 2). The role of HSF1 in cell/tissue/organism physiology is controversially discussed. Since most HSPs, particularly HSP70 and HSP90, are known to protect cells from deleterious effects of environmental stress, HSPs protect cells against stress-induced cell death. In cancer cells the protective function of HSPs and their master regulator, HSF1 contribute to tumor survival and metastatic dissemination. Thus, in the study by Dai et al. it was demonstrated that HSF1 expression was higher in the gastric cancer patients (as compared to normal tissue) and was correlated with poor overall survival and recurrence-free survival [27]. In the previous study, when cells with high and low malignant potential alongside their non-transformed counterparts were compared, the authors identified HSF1-regulated transcriptional program related to highly malignant cells that was distinct from heat shock response [28].

Multiple pro-cancer activities of HSF1 and HSP upregulation urged researchers to establish inhibitors [29,30] (Table 2).

One compound, triptolide, a diterpenoid triepoxide compound isolated from the Chinese herb *Tripterygium wilfordii*, has been shown to efficiently inhibit the HSPs synthesis [45]. The mechanisms of HSF1 inactivation by triptolide include inhibition of RNA polymerase [6] or targeting XPB, a subunit of the transcription factor TFIIH, which leads to the inhibition of RNA polymerase II-mediated transcription [46]. Triptolide has a high anti-tumor potency; it inhibited growth of pancreatic cancer cells in vitro already at a concentration of 50–200 nM. In in vivo models the drug reduced tumor growth when administered in a concentration 0.2 mg/kg/day for 60 days [47]. Similar effects of triptolide were demonstrated in cellular and animal models of neuroblastoma [48] and osteosarcoma [49]. Administration of triptolide reduced the MDM2 expression in human breast cancer cells and consequently, inhibited the activation of Akt. By regulating the MDM2/Akt pathway triptolide inhibited proliferation, induced apoptosis, and caused cell cycle arrest. Earlier the drug was demonstrated to activate caspase-mediated Bcl-2 cleavage, mitochondrial cytochrome c release and further activation of caspases [50].

Importantly, a combination of triptolide with doxorubicin administered to nude mice bearing breast cancer, inhibited tumor growth and demonstrated a higher anti-cancer effect than doxorubicin [31]. The clinical application of triptolide is limited by its high liver and kidney toxicity [51]. To overcome the toxicity problem, the compound was loaded together with curcumin into nanoparticles consisting of a mixture of mPEG-DPPE/calcium phosphate. This delivery system was found to synergistically increase pro-apoptotic and cell-arresting effects in SKOV-3 ovarian cancer cells when compared to the single drugs. The results of in vivo experiments showed that nanoparticles loaded by two drugs had no significant side effects but showed a synergistic anti-cancer effect [32]. It is still unclear, which activity of triptolide underlies its anti-cancer effects, however, reported studies on the role of HSF1 in tumor growth and maintenance indicated that a considerable part of the anti-cancer effects may be explained by triptolide-mediated inactivation of the heat shock response.

Another HSF1 inhibitor, benzylidene lactam compound, KNK-437, dose-dependently inhibited the development of thermotolerance and the induction of various HSPs including HSP105, HSP70, and HSP40 in COLO 320DM (human colon carcinoma) cells [52]. The compound sensitized bladder cancer cells to the proteasome inhibitor bortezomib (Velcade^®^) in a similar manner to the action of a sh-RNA-mediated silencing of the major stress-inducible member HSP70 (HSPA1A) [53]. The encouraging conception of simultaneous effects of proteotoxic stress combined with a suppression of the heat shock response through HSF1 was further exploited and extended by Bustany and co-authors who reported that blocking HSF1 with KNK-437 in combination with bortezomib exhibited additive effects on apoptosis induction in multiple myeloma cells from groups of patients with bad prognosis [54].

A compound termed KRIBB11 was found to inhibit the heat shock response by limiting HSF1-dependent recruitment of p-TEFb (positive transcription elongation factor b) to the HSP70 gene promoter. The substance promoted apoptosis and induced growth arrest in human colon HCT-116 cells [55]. In combination with benzimidazole carbamates, parbendazole and nocodazole, employed in treatment of colorectal cancers, KRIBB11 increased the latter potency two-fold probably by inactivating Erk1/2 signaling cascade [56]. More recently, KRIBB11 was used concurrently with the Akt inhibitor MK-2206 to reduce the rate of breast cancer metastasis. The application of this combination resulted in killing of cancer cells and breast cancer stem cells almost irrespective of their molecular subtypes. Furthermore, in a xenograft model of breast cancer a simultaneous targeting of Akt and HSF1 significantly reduced tumor growth, delayed outcome of metastasis, and prolonged the host survival [33]. The productive idea of concurrent and simultaneous inhibition of several signaling pathways including the chaperone system was tested in a study in which U251 glioma cells were treated with YM-1, a separator of the HSP70 complex with Bag-3 a co-chaperone, KRIBB11 as the inhibitor of a whole heat shock response and pan-Bcl-2 inhibitor AT-101. The drug complex reduced the HSP70, Bag-3, and the anti-apoptotic Bcl-2-like protein Mcl-1 and caused mitochondrial dysfunction resulting in apoptotic cell death via detachment from substrate [57]. However, Kang et al. demonstrated that KRIBB11 could accelerate Mcl-1 degradation via Mule-dependent pathway that is HSF-1 independent [58].

Importantly, KRIBB11 demonstrated beneficial results in inhibiting epithelial mesenchymal transition (EMT), particularly, reduced motility and invasion. These data were obtained in an orthotopic model of pancreatic cancer. Very similar results were reported when metformin, a known activator of AMP-dependent protein kinase, has been employed, and it was shown that this drug was also able to inhibit the HSF1 activity [59]. Although the authors have not applied the combination of KRIBB11 and metformin, one may suggest that it might provide therapeutic effects on highly aggressive tumors such as pancreatic ductal adenocarcinoma. The connection of metabolic and proteotoxic stress was demonstrated in experiments showing that an activation of AMPK by metformin which suppresses the HSF1 activity by phosphorylation of Ser121 inhibited tumor cell growth. Vice versa, proteotoxic stress inactivated AMPK and down-regulated the metabolic stress response [60].

In order to inhibit HSF1 in tumor cells with an extremely high HSP70 content, like glioblastoma, the protein kinase inhibitor D11 was applied together with 17-AAG, which is known to suppress the HSP90 activity. HSP90 inhibitors have been shown to induce resistance related to the increased synthesis of HSP70. The combination of both drugs was successful in detaching EGFR client proteins from HSP90 and to reduce the heat shock response which led to an enhanced cytotoxicity [61]. Furthermore, as was recently reported by Kühnel et al., combined therapeutic approach consisting of low concentrations of the HSP90 inhibitor NVP-AUY922 and knockdown of HSF1 using RNAi-Ready pSIRENRetroQ vectors significantly potentiated radiosensitization of tumor cells [62].

A deeper understanding of the mechanisms underlying the effects of anti-cancer drugs can unravel unexpected activities. Triptolide was previously thought to activate caspase-dependent apoptosis [50]. However, the cardenolide UNBS1450 which has cardiotonic activity, was found to inhibit synthesis of HSP70 both on mRNA and protein level. This indicates that the HSP70 reduction was regulated on the transcriptional activation of HSF1. In NCI-H727 and A549 NSCLC cells and in xenograft tumor mouse models a “non-classic” apoptosis signaling was induced by triptolide via the damage of lysosomal membranes [63].

After screening for low-toxic HSF1 inhibitors with the help of heat-shock-element-luciferase reporter assays we identified cardioglycoside CL-43 as a drug which was able to reduce the growth rate of cancer cells. Furthermore, we could show that CL-43 could elicit an enhanced efficacy of inefficient drugs like cisplatin in HCT-116 colon cancer cells [34].

In conclusion, the combinations of direct (KRIBB11) or indirect (triptolide, kinase inhibitors) HSF1 inhibitors with proteotoxic factors like proteasome inhibitors or traditional anti-cancer drugs, such as cytostatics, can overcome the resistance of tumor cells which is based on chaperones.

### 2.2. HSP90-Targeting Molecules in Anti-Cancer Therapeutic Schedules

The molecular chaperones of the HSP90 family in combination with their co-chaperones HSP70, HSP40, HiP and HoP play an important role in proteostasis, regulation of metabolic pathways and the protection of oncogenic factors that are involved in tumor progression and metastasis [64,65,66]. Like other chaperones, the functional activity of HSP90 is also dependent upon ATP. In case of ATP depletion, the HSP90 chaperone complex cannot mediate polypeptide homeostasis and therefore subsequently oncogenic client proteins undergo ubiquitin-mediated proteasomal degradation [67]. Application of various agents that interfere with the HSP90 chaperone cycle has emerged as a promising approach for targeting multiple oncogenic signaling pathways that are of high importance for tumor progression [68,69,70]. The employment of HSP90 inhibitors represents a plausible therapeutic strategy to target various cancer types with an overexpression of HSP90 [71,72,73,74] (Table 2). HSP90 inhibitors can be divided into two major groups: (i) inhibitors of the C-terminal domain (CTD) and (ii) inhibitors of the N-terminal ATPase domain (Figure 3).

Novobiocin, an antibiotic isolated from *Streptomyces*, represents one of the most potent inhibitors that binds to the CTD of HSP90 [73]. Several preclinical in vitro and in vivo studies demonstrated that novobiocin hampers the interaction of oncogenic client peptides and proteins with HSP90 and thus induces their ubiquitin-proteasomal mediated degradation [73,74]. However, due to the low therapeutic efficacy of the novobiocin several chemical modifications were subsequently proposed including chlorobiocin and coumermycin A1. Although these agents showed a promising therapeutic activity, none of these compounds have been assessed in clinical trials, yet [73,75]. Another inhibitor of the C-terminal domain, the primary flavonoid component of green tea—epigallocatechin gallate (EGCG)—was also shown to efficiently disrupt the dimerization of HSP90 [76,77]. However, subsequent studies demonstrated low availability of EGCG, poor metabolic and chemical stability that prevented further clinical trials of this compound.

Up-to-date the most studied inhibitors of the ATPase domain of HSP90 belong to the group of benzoquinone ansamycins (e.g., geldanamycin (GDA)) and antifungal macrolactone antibiotics (radicicol, RDC). As shown previously, GDA can efficiently block the phosphate region of the HSP90 binding pocket [77], whereas, RDC was demonstrated to bind to the ATPase domain of the chaperone that binds the adenine ring of ATP to produce hydrogen-bonding interactions within the protein binding pocket [78]. Subsequent studies have proven the therapeutic potency of GDA to reduce tumor progression. However, hydrophobicity and high hepatotoxicity significantly limit the application of the inhibitor in clinical practice. Therefore, novel HSP90 inhibitor derivatives with a lower hepatoxicity and an improved water solubility (i.e., 17-(allylamino)-17-demethoxy-geldanamycin (17-AAG) and 17-desmethoxy-17-*N,N*-dimethylaminoethylaminogeldanamycin (17-DMAG)) have been synthesized [79,80,81]. Modified inhibitors demonstrated an enhanced binding affinity and therapeutic efficacy as compared to GDA. Thus, 17-AAG IC_50_ values constituted 8–35 nM as compared to the IC_50_ values of GDA of μM range [77,82]. IC_50_ for 17-DMAG also ranged in nM values [83]. 17-AAG represents one of the best studied HSP90 inhibitors with anti-tumor efficacy in various cancer cell models and in subsequent clinical trials [35,84]. Although 17-AAG demonstrated a higher tumor-selective targeting compared to GDA, subsequent clinical phase I and II trials in breast cancer [85], metastatic pancreatic cancer [86], multiple myeloma [87], renal cell carcinoma [88], metastatic prostate cancer [89], and metastatic melanoma [90] failed to prove the therapeutic potential of the agent. Other synthesized small molecule inhibitors of HSP90, including AT13387, MPC3100, STA9090, XL888, NVP-AUY922, and purine-based compounds (PU-H71, CNF-2024 (BIIB021), PU-DZ8) have been investigated in clinical trials, however, they demonstrated only moderate clinical efficacy when applied as a monotherapy [91,92].

In contrast, when the inhibitor 17-AAG was employed in combination with trastuzumab in patients with HER2-positive metastatic breast cancer the authors reported an improved clinical outcome [93]. Indeed, a combination of the HSP90 inhibitors with conventional therapies (including chemo- and radiotherapies, targeted therapy, and immunotherapy) might significantly improve the therapeutic outcome [36,94,95]. Therefore, a combination of the chemotherapeutic agent 5-fluorouracil (5-FU) and anti-metabolites that impair DNA and RNA repair and synthesis, with HSP90 inhibitor ganetespib demonstrated a sensitizing effect in colorectal cancer cells (HCT-116, HT-29) and in a colorectal xenograft model [96]. Furthermore, a phase I clinical trial (NCT01226732) using AUY922 and capecitabine in patients with advanced solid tumors resulted in partial responses in 4/23 patients and stable diseases (median 25.5 weeks) in 8/23 patients [97]. Subsequent preclinical studies also have proven radiosensitization effects of HSP90 inhibitors in various tumor models [98,99].

Another therapeutic option for HSP90 inhibitors could be the combination with targeted therapies. The addition of the multi-kinase inhibitor sorafenib to tanespimycin demonstrated clinical efficacy in 4/6 melanoma and 9/12 renal cancer patients [100]. Due to the involvement of HSP90 in tumor angiogenesis several therapies are combining anti-angiogenic treatment with HSP90 inhibition. Although, the application of the ziv-aflibercept and ganetespib achieved a stable disease in 3/5 patients, the study had to be stoped due to severe adverse events [101].

Recent developments in the immunotherapy employing immune checkpoint inhibitors have led to the investigation of combinatorial approaches with HSP90 inhibitors. Mbofung et al. has shown an up-regulated therapeutic response to anti-CTLA4 and anti-PD-1 therapies in combination with the HSP90 inhibitor ganetespib in vivo due to an upregulation of the interferon response genes [36]. In another study a combination of the anti-PD-L1 antibody (STI-A1015) and ganetespib in the B16 melanoma and MC38 colon carcinoma models resulted in higher therapeutic efficacy as compared to the monotherapy regimens [37]. The effect of ganetespib could be explained partially by the influence of the HSP90 client proteins on the PD1 and PD-L1 expression as well as HIF-1α, JAK2 and mutated EGFR [37]. Another plausible mechanism of the anti-tumor activity of the HSP90 inhibitors is the increased presentation of oncogenic antigens upon proteasomal degradation. The application of 17-DMAG induced the degradation of EphA2 with a subsequent presentation of the tyrosine kinase receptor genes to EphA2-specific CD8+ T lymphocytes which caused protective anti-tumor immunity [102].

In conclusion, the application of HSP90 inhibitors in a series of preclinical studies demonstrated therapeutic potential, however, subsequent clinical trials reported only moderate effects when the reagent was used in a monotherapy. Presumably, low anti-cancer efficacy of HSP90 inhibitors could be explained by various specificity of the agents towards the four HSP90 paralogs which include two cytosolic forms (HSP90α (inducible/major form) and HSP90β (constitutive/minor forms), 94-kDa glucose-regulated protein (Grp94) in the ER (endoplasmic reticulum), and Trap1 (tumor necrosis factor receptor associated protein 1) in mitochondria [103]. Thus Liu et al. showed that NECA (5’-N-ethylcarboxamidoadenosine) inhibitor preferably targets cytosolic form of HSP90 with a less affinity towards Grp94 [104]. Combinatorial regimens with conventional treatment modalities (such as chemo- and/or radiotherapy), targeted therapy and immunotherapy could improve the anti-tumor activity of chaperone inhibitors suggesting further experimental and clinical studies.

### 2.3. Combinations of HSP70 Inhibitors with Anti-Tumor Drugs

Proteins belonging to HSP70 family are evolutionary highly conserved in their ATPase and substrate-binding domain and EEVD sequence at the C-terminus [105]. For their chaperone activity both domains are required, the ATPase domain serves as a receptor of ATP which is attached under the control of DNAJ class proteins and can be substituted with ADP in a reaction regulated by a few of nucleotide exchange factors, including HSP110, proteins of Bag family and others [106] (Figure 4). During the chaperone cycle the substrate, such as a newly synthesized polypeptide chains or stress-damaged proteins are captured by HSP70 and released refolded or become degraded by proteasomes via the HSP70/HSC70-Bag-CHIP complex. It is of importance that HSP70 and some of its co-chaperones, like Hdj1/HSP40, Bag-1 and Bag-3 are also over-expressed in many cancer types which recruit these proteins as a powerful anti-apoptotic machinery [107]. Bag-1 has a ubiquitin-like domain, which can target HSP70/Bag-1 complexes to proteasomal degradation [108], whereas Bag-3 represents a key player for an autophagic degradation pathway of client proteins in complex with HSPA8 [106,107]. There are numerous studies demonstrating an enhanced expression of HSP70 in a large variety of different tumor types. HSP70 is known to reduce aggregation of client proteins thus supporting the cellular proteostasis upon various stress conditions (i.e., oxidative stress, hypoxia, ionizing radiation, chemotherapy, etc). To cope with aggregation-prone polypeptides HSP70 together with its co-chaperones provides an attractive drug target. Upon drug treatment or radiotherapy the HSP70 synthesis is further enhanced. Many anti-cancer drugs trigger multiple signaling pathways that cause protein modifications, e.g., phosphorylation, dephosphorylation, acetylation or proteolysis. These events can change protein conformations which then serve as a target for HSP70. HSP70 exerts pleiotropic reactions in cancer cells which are prone to apoptosis. HSP70 binds and inactivates a number of pro-apoptotic molecules, i.e., caspase 3/7 and thereby delays apoptosis [107,108,109,110,111].

Up-to-date numerous HSP70 inhibitors were tested in various preclinical models, including MKT-077, PES (Pifithrin-μ), VER-155008, JG-98, Aptamer A-17, apoptozole, and others [112,113,114] (Table 2). These molecules bind to different parts of HSP70 with extremely high affinity and comparable IC_50_ values are reached in the nM range [115]. PES and PES-Cl recognize the substrate-binding domain (SBD); VER-155008 targets the ATP binding site, and MKT-077 links to an allosteric site near the ATP-binding site [116,117,118]. Recently, as was shown by Lazarev et al. colchicine derivative AEAE has been shown to target several distinct sites of the binding pocket of HSP70 with an average dissociation constant of nM [119]. A distinct group of compounds was shown to inhibit HSP70 association with its co-chaperones of DNAJ class, exemplified by myricetin [120] and MAL3-101 [121], or with Bag family of nucleotide exchange factors, like Thio-2 or JG-98 known to suppress binding of HSP70 to Bag-1 or Bag-3, respectively [110,122].

The main HSP70 inhibitor function separates client proteins from the chaperone similar to those shown for HSP90 inhibitors. In addition to its separation activity the HSP70 inhibitor JG-98 was found to destabilize its “client proteins” Akt1 and Raf1 in breast cancer cells MDA-MB-231 and MCF-7 [123]. BT-44 a benzodioxole derivative was found to disrupt the complex of HSP70 with the apoptosis inducer caspase 3. An etoposide-stimulated apoptosis in human U-937 leukemia cells could be recovered by BT-44 via dissociation of caspase 3 from HSP70 [124]. Erlotinib, an efficient inhibitor of EGFR tyrosine kinases is applied in the therapy of NSCLC harboring EGFR-activating mutations. A treatment with erlotinib inhibited the phosphorylation at tyrosine 41 and increases HSP70 ubiquitination with subsequent degradation of the protein. Intriguingly, induction of the HSP70 degradation enhanced the gene mutation rates in tumor cells indicating the role of the chaperone in the cell survival [125]. Combinations of HSP70 inhibitors and anti-cancer drugs are tested to date only at a pre-clinical level. It is not surprising that the most efficient drug complexes are using inhibitors targeting both chaperones, HSP70 and HSP90. One approach including VER-155008 and 17-AAG was found to efficiently eliminate NSCLC cells. VER-155008 was found to sensitize A549 cells to ionizing radiation [38]. In a more recent study HSP70 inhibitors, VER-155008 and MAL3-101 were tested either alone or concurrently with the HSP90 inhibitor, STA-9090, for their ability to reduce viability of muscle invasive bladder cancer cells. Combinations of VER-155008 with MAL3-101 synergistically lowered tumor cell viability while STA-9090+MAL3-101 also reduced cell viability but induced the expression of the cytoprotective HSP70 [39]. In another study VER-155008 was applied with the HSP90 inhibitor radicicol. This combination was found to be much more efficient in killing of anaplastic thyroid carcinoma cells than radicicol alone. The effect coincided with a decrease in the amount of HSC70 cognate protein, Akt and survivin which suggests the implication of multiple signaling cascades of apoptosis [40]. The rationale underlying the anti-chaperone drug combination effect may be based on the release of oncogenic client proteins from the chaperones which are important for cancer cell proliferation/survival and thereby allowing an improved apoptosis induction by cytostatic drugs. On the other hand, HSF1 is also among the client proteins of HSP90 and its release and subsequent activation causes an increased HSP70 synthesis which is cytoprotective. This unwanted effect must be neutralized by a co-administration of an HSP70 inhibitor.

One of the few proteotoxic factors approved for clinical use is local hyperthermia which besides positive anti-cancer effects causes an elevation of the expression of HSPs and an enhanced survival of tumor cells. This adverse effect can also be surmounted by HSP70 inhibitors. In two reports PES demonstrated a synergistic tumoricidal effect when applied to LNCaP and DU-145 human prostate cancer cells or administered before the application of magnetic fluid hyperthermia to HeyA8 intraperitoneal tumor model [126,127]. Proteotoxic stress which is a challenge for HSF1, HSP90 and also for HSP70 can be caused by inhibitors of proteasomes and autophagy. In a study performed to understand how to break HSP70-mediated protection rhabdomyosarcoma cells which are resistant to MAL3-101 chaperone inhibitor were genetically engineered. Notably, the resulting cells acquired activated endoplasmic reticulum-associated degradation pathways and an increased activity of autophagy. Chemical inhibitors (chloroquinone) or siRNA-mediated knock-down of the autophagy protein (ATG-5) restored MAL3-101 sensitivity and caused apoptosis [128].

A further interesting approach to increase the capacity of drug combinations with HSP70 inhibitors was recently demonstrated by Yaglom and coauthors. JG-98 was reported to dissociate the functional link between HSP70 and the co-chaperone Bag-3 to cause tumor cell death. A protein screening was performed to reveal efficient combinations of the inhibitor with other drugs. The search performed with Broad Institute Connectivity Map database and the IPAD service provided by ActivSignal, Inc. generated two pathways which are sensitive to concurrent treatment, proteasomes and RNApolII that were targeted by JG-98 concurrently with MG-132 and alpha-amanitin, respectively. Both drugs demonstrated synergistic anti-tumor activity in models of breast cancer [129].

Inhibition of HSP70 chaperone activity may lead to multiple defects in tumor cell signaling, particularly those involved in apoptosis. In one of such studies HSP70 inhibitor, pifithrin-μ was found to complement with the caspase-activating capacity of gambolgic acid leading to cumulative anti-tumor effect by targeting distinct pathways of apoptosis [130]. Pifithrin-μ was also employed in combination with cisplatin or oxaliplatin in the treatment of prostate PC-3 and colorectal HT-29 cancer cells and showed “moderate” and “significant” synergistic effects, respectively [41].

Another HSP70 inhibitor, PET-16, was employed to overcome the resistance of melanoma. In this study two polypeptides involved in enhanced tumorigenicity, phospho-FAK (PTK2) and mutant BRAF, were shown to be HSP70 client proteins. PET-16 synergized with BRAF inhibitor PLX4032 and reduced the level of phospho-FAK, impaired migration, invasion and metastasis in cell and animal models of melanoma [112].

Most HSP70 inhibitors were generated based on calculations of the efficacy of their molecule binding activity to HSP70 structure motifs e.g., with the use of molecular docking or molecular dynamics programs. Such approaches gave rise to YM-1, JG-98 [131] or to YK-5 chemical probe [116]. In our study the search of HSP70 inhibitors was focused on the compounds able to suppress two basic activities of HSP70, substrate-binding and refolding capacities. The appropriate assays were created and after the testing of more than 1000 compounds, 2–3 chemicals were found to inhibit the abovementioned activities of HSP70. One of these compounds, an amino-ethyl-amino-derivative of colchicine (AEAC) was found to bind the chaperone as proved by high-resolution assays including molecular docking and microscale thermophoresis (MST). Notably, AEAC was able to synergize with doxorubicin in killing mouse melanoma and rat glioblastoma cells in vitro and in vivo [119]. Interestingly, the toxicity of AEAC was lower by at least 20-fold than colchicine and the new compound had no microtubule-dissociating activity.

HSP70-binding peptides constitute distinct classes of HSP70 inhibitors. One of those, aptamer A17, expressing in human lung and breast cancer cells was demonstrated to increase the radiosensitizing effect of the HSP90 inhibitor NVP-AUY922, though the aptamer per se did not affect apoptosis [132]. The authors speculate that such effect may relate to the increase of DNA double strand breaks or up-regulated G2/M arrest caused by A17 expression. The aim of our study was to explore the effects of HSP70 peptides on the self-chaperone activity. According to numerous data certain peptide parts of the chaperone may interact with each other in the process of the protein oligomerization or with structurally similar and/or homologous peptides of the nucleotide-exchange factor HSP110. The data obtained with the aid of substrate-binding and refolding assays performed similarly to that in the search of AEAC (see above) allowed us to select the peptide showing high HSP70 binding capacities. Furthermore, the peptide ICyt2 significantly elevated the chemosensitivity of A431 epithelial carcinoma cells towards doxorubicin [124].

In conclusion, the inhibitors of HSP70 chaperone demonstrated a certain therapeutic potential that was further enhanced when the agents were combined with other treatment modalities. However, future preclinical studies should also include analysis of the inhibitors specificity towards HSP70 family homologous proteins, which currently include members in endoplasmatic reticulum and mitochondria as well as six members in the cytosol and nucleus. Low affinity of the inhibitor to various HSP70 homologs could result in the cancer cell resistance towards employed therapeutic strategies.

Apart from the inhibition of HSP70 the tumor-specific membrane expression of HSP70 can serve as a tumor target for anti-tumor immune responses [133,134] or for the specific nanoparticle-based therapies [135,136]. Our group has established a method to trigger the activity of natural killer (NK) cells by an HSP70 peptide TKD and low dose IL-2 to recognize and kill highly aggressive membrane HSP70 positive tumor cells [137]. Subsequent phase I clinical study demonstrated that treatment of tumor patients with autologous ex vivo HSP70-derived peptide plus IL-2 activated NK cells was safe and well tolerated [138]. The trial showed that intravenous administration of escalating numbers of treatment cycles (up to 6 cycles) of stimulated NK cells did not induce any severe toxicities. Clinical responses were assessed in patients with histologically confirmed metastatic colorectal cancer (*n* = 11) and non-small cell lung cancer (NSCLC) (*n* = 1). One stable disease was observed in one patient with colorectal cancer and one mixed response in NSCLC patient [138]. Presently the efficacy of TKD/IL-2 activated, autologous NK cells is tested in a randomized phase II clinical trial in patients with advanced NSCLC after standard radio-chemotherapy [138]. Patients (*n* = 90) with NSCLC in non-metastasized but locally advanced stages IIIA and IIIB after standard radio-chemotherapy (60–70 Gy; platinum based chemotherapy) will be enrolled and treated four times every 2–6 weeks with ex vivo TKD/IL-2 stimulated NK cells [139]. Subsequent preclinical studies further demonstrated that combination of activated NK cells with anti-PD-1 monoclonal antibodies resulted in tumor growth delay and increased overall animal survival in syngeneic GL261 glioblastoma or human xenograft A549 lung tumor models, indicating the therapeutic potency of adoptive cell therapies combined with immune check point inhibitors [140]. Indeed, recent case study of the patient with inoperable NSCLC (CT4, cN3, cM0, stage IIIb) treated with autologous ex vivo activated (TKD/IL-2) NK cells with anti-PD-1 antibody as a second-line therapy demonstrated a long-term tumor control [141].

### 2.4. Combinations of HSP27 Inhibitors with Anti-Tumor Drugs

HSP27 (HSPB1) belongs to a separate group of so-called small heat shock proteins. Like many other chaperones it demonstrates cytoprotective activities. Its chaperone activity is induced by the phosphorylation and thereby HSP27 multimers prevent aggregation and/or regulate activity and degradation of certain client proteins. HSP27 expression becomes highly up-regulated in cancer cells after chemotherapy indicating that the chaperone impacts on tumor cell resistance and progression in bladder, lung and other types of cancer [142]. It is also of importance that HSP27 promotes interleukin-6-mediated EMT in prostate cancer cells via modulation of STAT3/Twist signaling [143]. Among the HSP27 inhibitors most notable is OGX-427, an anti-sense oligonucleotide (apatorsen) (Table 2). The efficacy of the oligonucleotide was proven in a variety of animal cancer models and a few years ago it was studied in phase II clinical trials [144]. High therapeutic activity of OGX-427 was shown in a few other anti-cancer combinations, particularly with traditional drugs, such as gemicatabine [145] or docetaxel [42] in phase II clinical trials. As in the case of other chaperone inhibitors their combinations with proteotoxic factors showed a remarkable efficacy. In one of the studies OGX-427 was employed concurrently with inhibitor of autophagy chloroquinone which significantly inhibited prostate tumor growth in animal models [43]. Also similar to concurrent inhibition of two major chaperones (see the description of combination of HSP70/HSP90 inhibitors in this section) administration of Hsp90 inhibitor PF-04928473 with OGX-427 was found to efficiently suppress tumor cell growth and induce apoptosis. In a xenograft castrate-resistant prostate cancer model the above mentioned combination caused an enhanced delay of tumor growth and a prolonged survival of animals [44].

## 3. Conclusions

Heat shock proteins have been demonstrated to play key roles in tumor progression and resistance to currently applied therapies and thus could be employed as a potential target in development of new therapeutic approaches. However, recent clinical trials using established inhibitors against HSP90 or HSP70 demonstrated limited clinical efficacy and undesired toxicity when being employed as a monotherapy. Presumably, combinations of the inhibitors with standard chemotherapeutic agents or targeted therapies might improve the anti-tumor potency of the HSP-inhibitors even at lower concentrations and thereby reduce their side effects. Another approach is based on combinations of several HSP inhibitors in cancer therapy. A combination of HSP90 and HSP70 inhibitors could revert the compensatory effects of HSP90 inhibitors towards an enhanced expression of HSP70 in cancer cells. Novel approaches using different HSP-inhibitors in combination with conventional therapeutic strategies might provide promising strategies to improve clinical outcome of therapy-resistant cancers.

## Figures and Tables

**Figure 1 ijms-20-05284-f001:**
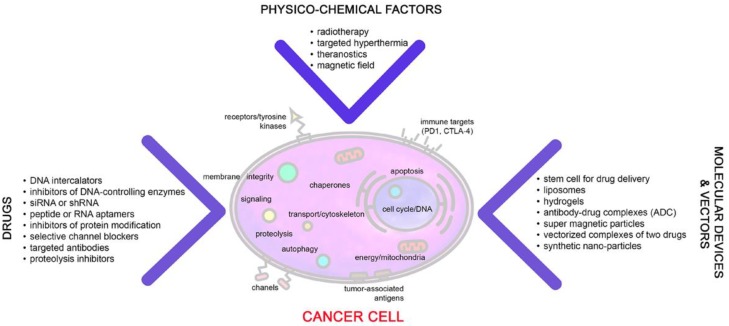
Currently used drugs, physico-chemical factors, molecular devices and vectors, which are presently tested in combinatorial treatment approaches of cancer.

**Figure 2 ijms-20-05284-f002:**
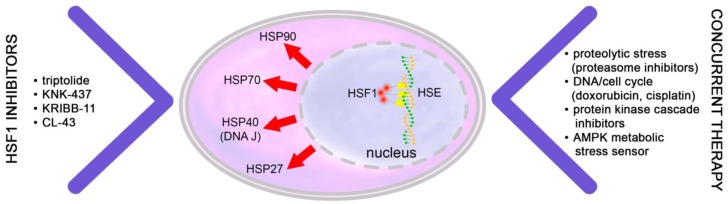
Inactivation of HSF1 transcription activator synergizes with anti-cancer therapy. Activation of HSF1 leads to elevated synthesis of the set of protective heat shock proteins while inhibition of the regulator (shown at the left side) combined with cytotoxic agents (right column) leads to enhanced tumor cell death.

**Figure 3 ijms-20-05284-f003:**
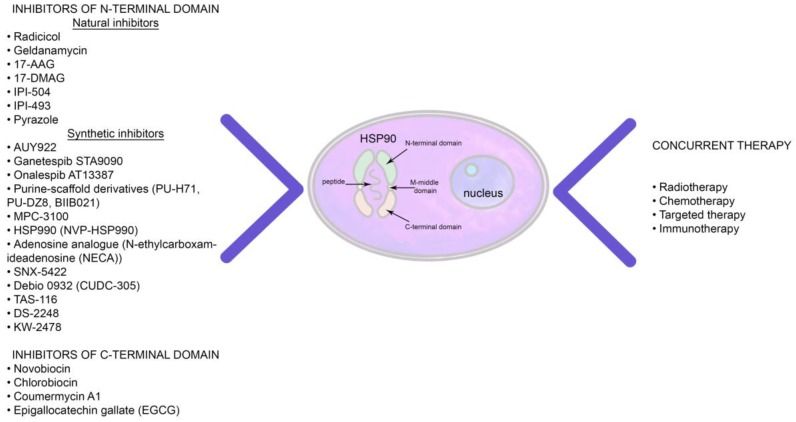
Concurrent application of the HSP90 inhibitors and conventional therapeutic approaches.

**Figure 4 ijms-20-05284-f004:**
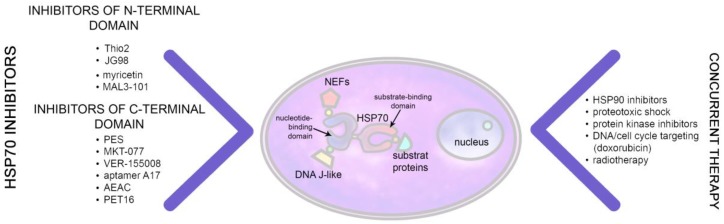
Concurrent application of HSP70 inhibitors and anti-tumor therapies.

**Table 1 ijms-20-05284-t001:** Combinatorial anti-tumor therapies.

Therapeutic Factors 1	Factor 2	Cancer Type	Outcome	Reference
AC220 quizartinib (FLT3 TK inhibitor)	TAK-165 HER2 inhibitor	Variety of human tumors	Cytotoxicity, autophagy	Ouchida et al., 2018 [11]
Vemurafenib, inhibitor of mutant BRAF	PAC-1 pro-caspase activator	Human melanoma in vitro & in vivo	Effect in caspase-3 activation, inhibition of tumor regrowth	Peh et al., 2016 [12]
Metformin AMPK activator	Cisplatin	Skov-3 and HEY ovarian cancer in vitro & in vivo	Inhibition of T and Smad-Smad3 phosphorylation GFβ1 expression	Zheng et al., 2018 [13]
Cisplatin packed together in amphiphilic PLG-g	Do Ve/PEG graft copolymer cetaxel	B16F1 cells & graft	Anti-metastasis effect and prolonged circulation	Song et al., 2014 [14]
Cetuximab anti-ErbB/HER monoclonal antibody	Erlotinib tyrosine-kinase inhibitor	NSCLC non-small-cell lung cancer	Antibody-dependent, NK mediated cytotoxicity	Cavazzoni et al., 2012 [15]
Anti-KRAS antibody	Gemcitabine	Pancreatic cancer in vitro & in vivo	Inhibition RAS signaling	Kang et al., 2018 [16]
Bevacizumab CTLA4 blockade	Ipilimumab VEGF inhibition	Patients with metastatic melanoma	Survival up to 25 months, immune response	Hodi et al., 2014 [17]
Trametinib and dabrafenib	Anti-PD1 antibody	BRAF(V600E) melanoma	Anti-tumor effect in vivo, reduction of metastasis	Hu-Lieskovan et al., 2015 [18]
Dabrafenib or trametinib BRAF & MEK inhibitors	Anti-PD-1, PD-L1, and CTLA-4 (checkpoints) antibodies	Carcinoma in vitro & in vivo	Anti-tumor immune response	Liu et al., 2015 [19]
Physical exercise or dihydroartemisinin (inducer of oxidative stress)	Temozolomide	Glioblastoma in vitro & in vivo	Reduced clonogenicity/migration, lowered metastasis	Lemke et al., 2016 [20]
Oncolytic virus both encapsulated in extracellular vesicles	Paclitaxel	Lung cancer	Anti-tumor effect in vivo	Garofalo et al., 2018 [21]

**Table 2 ijms-20-05284-t002:** Concurrent anti-tumor therapies employing inhibitors of HSPs.

Inhibitor	Concurrent Therapy	Cancer Type	Outcome	Reference
**HSF1 inhibitors**				
Triptolide	Doxorubicin	MCF-7 and MDA-MB-468 human breast cancer	Inhibition of tumor growth and enhancement of anti-tumor effects of doxorubicin	Xiong et al., 2016 [31]
Triptolide	Curcumin	Ovarian cancer	Tumor inhibition rate of 68.78%	Liu et al., 2018 [32]
KRIBB11	Akt small molecule inhibitor MK-2206	Breast cancer	Synergistic killing of breast cancer cells and breast cancer stem cells; inhibition of tumor growth	Carpenter et al., 2017 [33]
Cardenolide CL-43	Cisplatin/etoposide/doxorubicin	HCT-116 human colon carcinoma	Additive anti-tumor effect	Nikotina et al., 2018 [34]
**HSP90 inhibitors**				
Tanespimycin (17-AAG)	Trastuzumab	HER2-positive metastatic breast cancer progressing on trastuzumab	Significant anticancer activity	Modi et al., 2011 [35]
Ganetespib (STA-9090)	BRAF(V600E) inhibitor vemurafenib/MEK inhibitor TAK-733	Melanoma	Tumor regression in vemurafenib-resistant xenografts	Acquaviva et al., 2014 [36]
Ganetespib	Anti-PD-L1 antibody STI-A1015	MC38 colon carcinoma and B16 melanoma	Enhanced anti-tumor efficacy of the combinatorial regimen	Proia et al., 2015 [37]
**HSP70 inhibitors**				
VER-155008	17-AAD inhibitor of HSP90	NSCLC cells	Synergistic effect on NSCLC cells proliferation	Wen et al., 2014 [38]
VER-155008/MAL3-101	STA-9090 inhibitor of HSP90	Muscle invasive bladder cancer (MIBC) cells	Synergistic anti-tumor effect	Prince et al., 2018 [39]
VER-155008	Radicicol inhibitor of HSP90	Anaplastic thyroid carcinoma cells	Enhanced anti-tumor activity of combinatorial therapy	Kim et al., 2014 [40]
Pifithrin-μ	Cisplatin/Oxaliplatin	HT29 colorectal and PC-3 prostate cancer cells	Synergistic anti-tumor effect	McKeon et al., 2016 [41]
**HSP27 inhibitors**				
Apartorsen	Docetaxel	Platinum-resistant metastatic urothelial carcinoma	Improved OS compared to docetaxel alone	Rosenberg et al., 2018 [42]
OGX-427	Autophagy inhibitor chloroquine	PC-3 prostate cancer	Inhibition of tumor progression in vivo	Kumano et al., 2012 [43]
OGX-427	HSP90 inhibitor PF-04929113	Castrate-resistant prostate cancer	OGX-427 synergistically enhanced anti-tumor effect of HSP90 inhibitor	Lamoureux et al., 2014 [44]

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
