# Peer review of "Combination of Anti-Cancer Drugs with Molecular Chaperone Inhibitors"

_ijms, 2019, doi:10.3390/ijms20215284_

Round 1

Reviewer 1 Report

The review covers the use of inhibitors of the heat shock response and chaperones in the potential treatment of cancer. A particular focus is on combinations of drugs and other treatments. The manuscript is divided into parts on the general use of combination therapies, and inhibitors of Hsf1, Hsp90, Hsp70 and Hsp27. It is a very active field of research and the review is timely, and will be of interest to the audience. Coverage of the literature is enough to be useful.

The main concern is with the writing. There are a number of mis-statements, and parts of the manuscript are disorganized. I recommend that it also be checked for grammar.

Specific points:

Page 2, line 54: nucleic acid

Page 2, lines 67-68 describes packaging in nanoparticles, this belongs later with p. 4, lines 86-92 where other packaging approaches are described.

Page 3, line 83. The target of metformin is controversial. There are effects which appear to be independent of AMPK, and perhaps directly on mitochondrial respiration. It is possible the drug effects are due to multiple modes of action. The combination with cisplatin, which is even more non-specific, is not a good example compared to the other specific compounds mentioned.

Page 4, line 134: It is now well established that the Hsp110 family are homologs of Hsp70 so they do not need to be counted as a separate family. Indeed, Hsp110 is already left out of Fig. 2. The preferred names for the Hsp40 family are the J-domain proteins or DNAJ, as in the figure, and should be used in the text.

Lines 139-140: Hsps levels... aggregates after oxidative stress, reference 25 Dai et al. does not address oxidative stress and aggregation at all. It is an observational study that correlates high HSF1 expression with gastric cancer prognosis. The statement on aggregates is speculative.

Lines 140-142: “the expression of genes belonging to distinct of Hsps signaling systems” What is meant here? The Mendillo et al. paper reference 26 is very important, because it shows a role of HSF1 in cancer that is unrelated to the expression of Hsps. The point should be made very clearly. It does not invalidate the important of Hsps in many cancers.

Lines 170-172: “recent data on the role of Hsf1” no references provided. There are many effects of triptolide which are not related to Hsf1, and the direct targets of triptolide are XPB and possibly other proteins which are not Hsf1. The statement on lines 148-149 is potentially misleading, as it indicates triptolide is a specific inhibitor of Hsf1. I suggest the wording be softened: “Multiple pro-cancer activities of Hsf1 and Hsp upregulation urged researchers to establish inhibitors. One compound/drug...” The manuscript recognizes later on that the only compound that directly targets Hsf1 is KRIBB11.

Lines 193-198: The loss of Mcl-1 in Antonietti et al. 2017, may be due to an HSF1 independent effect on the Mule E3 ubiquitin ligase which regulates Mcl-1, Kang et al. 2017 Biochem Biophys Res Comm.

Line 256, chemically, radicicol is not a benzoquinone ansamycin like GDA or herbimycin, although it targets the same site on Hsp90.

Lines 317-319, the Hsp70/Hsc70-Bag-CHIP complex... Hdj1/Hsp40 and Bag-1-Bag-3: Degradation is primarily through Hsp70/Hsc70-CHIP. Bag1 may promote proteasomal degradation through different mechanisms. Bag3 is thought to promote autophagy in opposition to proteasomal degradation, in complexes with Hsp70, CHIP and HspB8. The way it is written it could suggest Bag1 and Bag3 form a complex or are different names for the same protein like Hdj1 and Hsp40, or close homologs like Hsp70/Hsc70. References to the original literature or a review on mechanism, would be preferred to the Calderwood & Gong 2016 review, which is focused on cancer.

Pages 9-11: The section on Hsp70 inhibitors is very poorly organized, in contrast to the Hsf1 and Hsp90 inhibitors. There is almost no mention of drug mechanisms even when they are known. JG-98 is introduced at different places, and it is not explained that PES is the same as pifithrin-μ, or that PET-16 is its derivative. I suggest it be re-written by grouping all descriptions of a drug or drug family together and introducing their mechanism and target site on Hsp70. As an example, VER-155008 competes ATP from Hsp70... MAL3-101 interferes with the activation of the Hsp70 ATPase... JG-98 targets the interaction between Hsp70 and Bag3, which has the result of separating clients from the chaperone... and so on.

Page 11: It is a personal preference, but a short concluding statement on future prospects would be a better way to end the review, instead of just stopping after Hsp27.

Author Response

We are grateful for the provided comments and propose the revision of our manuscript. According to the Editorial office comments we have revised our manuscript and have incorporated the changes.

Reviewer 1

The review covers the use of inhibitors of the heat shock response and chaperones in the potential treatment of cancer. A particular focus is on combinations of drugs and other treatments. The manuscript is divided into parts on the general use of combination therapies, and inhibitors of Hsf1, Hsp90, Hsp70 and Hsp27. It is a very active field of research and the review is timely, and will be of interest to the audience. Coverage of the literature is enough to be useful.

The main concern is with the writing. There are a number of mis-statements, and parts of the manuscript are disorganized. I recommend that it also be checked for grammar.

Specific points:

Page 2, line 54: nucleic acid

ANSWER: This was corrected. The Ms was revised accordingly.

Page 2, lines 67-68 describes packaging in nanoparticles, this belongs later with p. 4, lines 86-92 where other packaging approaches are described.

ANSWER: This was corrected.

Page 3, line 83. The target of metformin is controversial. There are effects which appear to be independent of AMPK, and perhaps directly on mitochondrial respiration. It is possible the drug effects are due to multiple modes of action. The combination with cisplatin, which is even more non-specific, is not a good example compared to the other specific compounds mentioned.

ANSWER: We totally agree with the reviewer. We have deleted this part of the manuscript.

Page 4, line 134: It is now well established that the Hsp110 family are homologs of Hsp70 so they do not need to be counted as a separate family. Indeed, Hsp110 is already left out of Fig. 2. The preferred names for the Hsp40 family are the J-domain proteins or DNAJ, as in the figure, and should be used in the text.

ANSWER: We have corrected accordingly.

Lines 139-140: Hsps levels... aggregates after oxidative stress, reference 25 Dai et al. does not address oxidative stress and aggregation at all. It is an observational study that correlates high HSF1 expression with gastric cancer prognosis. The statement on aggregates is speculative.

ANSWER: We agree with the reviewer and have accordingly rewritten this paragraph in the manuscript.

Lines 140-142: “the expression of genes belonging to distinct of Hsps signaling systems” What is meant here? The Mendillo et al. paper reference 26 is very important, because it shows a role of HSF1 in cancer that is unrelated to the expression of Hsps. The point should be made very clearly. It does not invalidate the important of Hsps in many cancers.

ANSWER: We agree with a reviewer and have rewritten this paragraph.

Lines 170-172: “recent data on the role of Hsf1” no references provided. There are many effects of triptolide which are not related to Hsf1, and the direct targets of triptolide are XPB and possibly other proteins which are not Hsf1. The statement on lines 148-149 is potentially misleading, as it indicates triptolide is a specific inhibitor of Hsf1. I suggest the wording be softened: “Multiple pro-cancer activities of Hsf1 and Hsp upregulation urged researchers to establish inhibitors. One compound/drug...” The manuscript recognizes later on that the only compound that directly targets Hsf1 is KRIBB11.

ANSWER: The Ms has been corrected according to the reviewers comment.

Lines 193-198: The loss of Mcl-1 in Antonietti et al. 2017, may be due to an HSF1 independent effect on the Mule E3 ubiquitin ligase which regulates Mcl-1, Kang et al. 2017 Biochem Biophys Res Comm.

ANSWER: We have added these comments.

Line 256, chemically, radicicol is not a benzoquinone ansamycin like GDA or herbimycin, although it targets the same site on Hsp90.

ANSWER: We agree with a reviewer and have corrected this sentence.

Lines 317-319, the Hsp70/Hsc70-Bag-CHIP complex... Hdj1/Hsp40 and Bag-1-Bag-3: Degradation is primarily through Hsp70/Hsc70-CHIP. Bag1 may promote proteasomal degradation through different mechanisms. Bag3 is thought to promote autophagy in opposition to proteasomal degradation, in complexes with Hsp70, CHIP and HspB8. The way it is written it could suggest Bag1 and Bag3 form a complex or are different names for the same protein like Hdj1 and Hsp40, or close homologs like Hsp70/Hsc70. References to the original literature or a review on mechanism, would be preferred to the Calderwood & Gong 2016 review, which is focused on cancer.

ANSWER: We have modified this paragraph and added the references

Pages 9-11: The section on Hsp70 inhibitors is very poorly organized, in contrast to the Hsf1 and Hsp90 inhibitors. There is almost no mention of drug mechanisms even when they are known. JG-98 is introduced at different places, and it is not explained that PES is the same as pifithrin-μ, or that PET-16 is its derivative. I suggest it be re-written by grouping all descriptions of a drug or drug family together and introducing their mechanism and target site on Hsp70. As an example, VER-155008 competes ATP from Hsp70... MAL3-101 interferes with the activation of the Hsp70 ATPase... JG-98 targets the interaction between Hsp70 and Bag3, which has the result of separating clients from the chaperone... and so on.

ANSWER: We have modified the section concerning the Hsp70 inhibitors.

Page 11: It is a personal preference, but a short concluding statement on future prospects would be a better way to end the review, instead of just stopping after Hsp27.

ANSWER: We agree with a reviewer. We added section ‘3. Conclusion’ section to the manuscript.

Reviewer 2 Report

The submitted review manuscript entitled “Combination of anti-cancer drugs with molecular chaperone inhibitors” by  Shevtso et al., describes the therapeutic potential of combining inhibitors of molecular chaperones with anti-cancer treatments. The manuscript is a very thorough and timely review of an emerging literature, especially given the recent clinical trials in this area. The use of chaperone inhibitors with other drugs/treatment modalities is well-rationalized from a mechanistic standpoint.  Such a review will appeal to a wide audience including cancer biologists, oncologists, pharmacologists and those who study heat shock proteins and proteostasis.  

There are several points to consider for improving the manuscript:

The title only reflects that the review will cover the combination of molecular chaperone inhibitors with other drugs and not the various other treatment modalities. Perhaps it would be good to modify the title to represent the broader scope of the review.

The lengthy “overview” section distracts from the molecular chaperone focus of the review. The general utility of combination treatments as therapeutically valid could be conveyed in less detail, perhaps in an opening paragraph with citation of other reviews. Along these same lines, while interesting Table 1 is a distraction.  

Writing: In some parts of the manuscript the writing is awkward in word usage (Ex: line 320 “tumor entities”) and sentence construction (Ex: sentence beginning on line 320; “Hsp70 is known to reduce aggregation of damaged proteins and this action extends to numerous harmful conditions, ……products of tumor degradation”. Perhaps an editorial service would be useful.

Regarding the use of the term synergistic (or variants thereof) there are 33 instances of “synerg” in the manuscript. Are the papers that are being cited for reporting synergistic effects actually using this terminology and if so, how was that determined?  Typically, isobologram analysis is performed to distinguish drug synergy vs drug additivity (http://jpet.aspetjournals.org/content/319/1/1).   If the effects being referred to are only “additive” the correct terminology needs to be applied.  These terms need to be defined.

A table that focuses on combinations of chaperone inhibitors with other treatment modalities would be very useful (much like table 1 but focused on chaperones).

Line 341 “The reduction in the Hsp70 content is the most likely reason for an increased DNA damage and enhanced gene mutation rate in the tumor cells [107].” Please reconcile this notion with the anti-cancer effects of Hsp70 inhibitors.

It would be of interest for the reader to have a little more detail on the clinical trials references in lines 427+.

A concluding paragraph needs to be added.

Author Response

We are grateful for the provided comments and propose the revision of our manuscript. According to the Editorial office comments we have revised our manuscript and have incorporated the changes.

The submitted review manuscript entitled “Combination of anti-cancer drugs with molecular chaperone inhibitors” by  Shevtsov et al., describes the therapeutic potential of combining inhibitors of molecular chaperones with anti-cancer treatments. The manuscript is a very thorough and timely review of an emerging literature, especially given the recent clinical trials in this area. The use of chaperone inhibitors with other drugs/treatment modalities is well-rationalized from a mechanistic standpoint.  Such a review will appeal to a wide audience including cancer biologists, oncologists, pharmacologists and those who study heat shock proteins and proteostasis.  

There are several points to consider for improving the manuscript:

The title only reflects that the review will cover the combination of molecular chaperone inhibitors with other drugs and not the various other treatment modalities. Perhaps it would be good to modify the title to represent the broader scope of the review.

ANSWER: Although the major part of the manuscript refers to the combination therapy of HSPs inhibitors with other drugs (i.e., targeted therapy, chemotherapy, etc) we also included data on the combination of inhibitors with other therapies (e.g., radiotherapy, thermotherapy, etc).

The lengthy “overview” section distracts from the molecular chaperone focus of the review. The general utility of combination treatments as therapeutically valid could be conveyed in less detail, perhaps in an opening paragraph with citation of other reviews. Along these same lines, while interesting Table 1 is a distraction.  

ANSWER: Table 1 represents the examples of concurrent therapies in translational and clinical oncology. We added Table 2 to the current revision version that demonstrates the examples of HSPs inhibitors with other treatment modalities.

Writing: In some parts of the manuscript the writing is awkward in word usage (Ex: line 320 “tumor entities”) and sentence construction (Ex: sentence beginning on line 320; “Hsp70 is known to reduce aggregation of damaged proteins and this action extends to numerous harmful conditions, ……products of tumor degradation”. Perhaps an editorial service would be useful.

ANSWER: We corrected the manuscript accordingly.

Regarding the use of the term synergistic (or variants thereof) there are 33 instances of “synerg” in the manuscript. Are the papers that are being cited for reporting synergistic effects actually using this terminology and if so, how was that determined?  Typically, isobologram analysis is performed to distinguish drug synergy vs drug additivity (http://jpet.aspetjournals.org/content/319/1/1).   If the effects being referred to are only “additive” the correct terminology needs to be applied.  These terms need to be defined.

ANSWER: We thank the reviewer for this notion. We have corrected the manuscript accordingly throughout the text.

A table that focuses on combinations of chaperone inhibitors with other treatment modalities would be very useful (much like table 1 but focused on chaperones).

ANSWER: We have added Table 2 that shows concurrent therapies of HSPs inhibitors and other agents.

Line 341 “The reduction in the Hsp70 content is the most likely reason for an increased DNA damage and enhanced gene mutation rate in the tumor cells [107].” Please reconcile this notion with the anti-cancer effects of Hsp70 inhibitors.

ANSWER: We have corrected this.

It would be of interest for the reader to have a little more detail on the clinical trials references in lines 427+.

ANSWER: We have expanded this section.

A concluding paragraph needs to be added.

ASNWER: We have added ‘3. Conclusion’ section to the manuscript.

Reviewer 3 Report

Summary

            The authors have set out here to describe the current literature regarding available inhibitors of heat shock response as either a single treatment or combination therapies. From this, it is clear that synergistic effects are commonly observed, which thereby allow for lowered doses and decreased subsequent toxicity. Emphasis is placed on such treatments targeting Hsf1, Hsp90, Hsp70, and Hsp27.

Comments

Overall, the manuscript thoroughly addresses the effects of inhibitor administration either by single or combination treatment. However, the authors often fail to provide broader context. That is, the recurring pattern is to describe the inhibitor and the downstream effect, but not much interpretation is provided. In order to increase accessibility to the reader, more background material needs to be included to aid in reader comprehension. Given the emphasis on Hsp90,70, and 27, more background needs to be included to compare and contrast these enzymes. Furthermore, an overview of the architectural arrangements needs to be included since direct references are made to Hsp domains without any introduction. Some effort should also be made to describe the nucleotide binding sites since these are explicitly referred to in the context of where inhibitors bind. For example, pg. 23 “…GDA can efficiently block the phosphate region of the Hsp90 binding pocket…” However, no effort is currently made to describe the “phosphate region.” A structural representation of the nucleotide binding site(s) would help. When discussing inhibitor potencies, a description of estimated binding affinities, IC50, EC50, etc. should be provided to allow for comparison of inhibitors. Given that a wide range of inhibitor molecules are discussed, one addition that would improve the impact of the manuscript would be a discussion of common interaction modes shared amongst inhibitors. Can they be grouped based on common interactions? For example, does one class of inhibitor interact with the ATP binding site (or particular elements therein) and another group with a separate Hsp domain?

Author Response

We are grateful for the provided comments and propose the revision of our manuscript. According to the Editorial office comments we have revised our manuscript and have incorporated the changes.

Summary

            The authors have set out here to describe the current literature regarding available inhibitors of heat shock response as either a single treatment or combination therapies. From this, it is clear that synergistic effects are commonly observed, which thereby allow for lowered doses and decreased subsequent toxicity. Emphasis is placed on such treatments targeting Hsf1, Hsp90, Hsp70, and Hsp27.

Comments

Overall, the manuscript thoroughly addresses the effects of inhibitor administration either by single or combination treatment. However, the authors often fail to provide broader context. That is, the recurring pattern is to describe the inhibitor and the downstream effect, but not much interpretation is provided. In order to increase accessibility to the reader, more background material needs to be included to aid in reader comprehension. Given the emphasis on Hsp90, 70, and 27, more background needs to be included to compare and contrast these enzymes. Furthermore, an overview of the architectural arrangements needs to be included since direct references are made to Hsp domains without any introduction. Some effort should also be made to describe the nucleotide binding sites since these are explicitly referred to in the context of where inhibitors bind. For example, pg. 23 “…GDA can efficiently block the phosphate region of the Hsp90 binding pocket…” However, no effort is currently made to describe the “phosphate region.” A structural representation of the nucleotide binding site(s) would help. When discussing inhibitor potencies, a description of estimated binding affinities, IC50, EC50, etc. should be provided to allow for comparison of inhibitors. Given that a wide range of inhibitor molecules are discussed, one addition that would improve the impact of the manuscript would be a discussion of common interaction modes shared amongst inhibitors. Can they be grouped based on common interactions? For example, does one class of inhibitor interact with the ATP binding site (or particular elements therein) and another group with a separate Hsp domain?

ANSWER: We would like to thank the reviewer for the provided comments. We have substantially revised the manuscript. Thus we added ‘3. Conclusion’ section and revised all parts of the text. Thus we have restructured the section concerning Hsp70 inhibitors classifying the agents according the binding site of the protein. Additionally, we added IC50 for certain described inhibitors. The added a new Table 2 that describes examples of concurrent therapies employing HSPs inhibitors and other agents.

Reviewer 4 Report

This manuscript has the aim to review the literature on cancer treatment with a combination of molecular chaperone inhibitors and anti cancer drugs that address other targets used to counteract tumor growth and cancer cell survival. Most of the clinical, basal in vitro and in vivo studies discussed focus on heat shock factor 1, which is involved in stress-induced expression of heat shock and stress response proteins and the two chaperone systems HSP70 and HSP90. While the manuscript presents an ample record of the relevant literature, it is difficult to read as it lacks a clear structure and is superficial in describing mechanisms. The basic structure with the 2 headings and 4 subheadings is fine, but the text should focus more on what the abstract promises to deal with and be shortened combinatorial treatments involving chaperone-related drugs. Maybe a 3rd main heading summarizing/concluding at the end would also support readability and clarity. The many acronyms, short terms for cancer types and many terms just used once or twice are a challenge for the reader; some simplification, use of more descriptive terms, and avoiding using different terms for the same things would improve readability.

There are many long and nested sentences, some in which the syntax appears odd. I am not a native speaker, but I think review by a native speaker and addition of commas would be helpful.

Wording: The synonymous use of the terms’ combinational’, ‘combined’ and ’combinatorial’ is a bit confusing.

Overview

The first section is supposed to give an overview on combinatorial anti-cancer therapies. It starts with a general rationale for the advantages of combining drugs that target different cancer mechanisms. But it then follows tangents giving examples on which types of molecules are targeted, comments websites where one can look up the targets and effects of a given drug, and considerations and new techniques to deliver drugs more efficiently and increase stability. Before coming to the point, combining chaperoning mechanisms in combination with other targets, there is a lengthy and unconcise discussion of other targets. The first part of the last paragraph in section 1, without the list of examples, might instead be used to start this section. Then, continue shorter and more focused towards treatments with chaperone effectors plus another target.

The mechanistic rationale for targeting molecular chaperones and the difference of this target type from other anti-cancer therapies is not well defined. The same holds true for the introduction of misfolding protein aggregation, proteotoxic stress, and proteostasis.

Table 1 is in its current form not very helpful. As the focus is on chaperone plus other target drug, a table on such combinations would be better. Or have a table of combinatorial treatments with key instances where no chaperone-targeting therapy was used and a clearly separated second part or a separate table 2 with key examples for the HSF1, HSP90, HSP70, or HSP27 targeting reports. A simplified keyword for the targeted mechanism would also be helpful. The table(s) should also in the text be described as for example listing selected examples, and what the criteria for selection (different types, most successful, or else) were applied.

Specific comments to this section:

Lines 52-55: At the cellular level the drugs target mechanisms of protein modifications like phosphorylation, proteolysis or acetylation, while among the most desirable cellular targets are mitochondrial functions, membrane integrity, nuclear acid-protein complexes, cell transport mechanism, proteolytic machinery and others (Fig. 1). What is meant by desirable and how does this contrast to the mechanisms listed in the first part of the sentence?

Lines 82-85: Similarly, metformin, an activator of AMP-dependent protein kinase metabolic stress machinery and commonly employed cisplatin notably reduced metastasis by affecting the epithelial-mesenchymal transition (EMT) in in vivo ovarian cancer models [13]. Activation of AMPK is just one of many reported effects of metformin; naming it here alone and without mentioning the target for cisplatin is insufficient.

HSF1

As stated in the abstract of the quoted reference (24), HSF1 is involved in regulation of heat shock and stress proteins, but also many functions and mechanisms that are not directly related to the accumulation of misfolding and aggregating proteins. Furthermore, not all HSPs are true heat shock regulated proteins. Especially for the HSP70 family, constitutively and stress regulated orthologs exist.

The description of pro-tumoric effects of HSF1 and the rationale for the finding that triptolide is ‘specific’ is not well understandable. It ‘has been shown to efficiently inhibit the Hsp synthesis’. Does this mean synthesis of all/many proteins whose genes possess HSE’s? Why is this more specific than inhibiting HSF1 in other ways?

HSP90

There are 4 human HSP90 orthologs, two cytosolic, one in the ER and one in mitochondria. Do the statements apply for all or only for the cytosolic ones that have the function to modify the conformations of transcription factors. HSP90 activity inhibitors likely affect them all. I would not consider HSP70 a co-chaperone of HSP90, rather a chaperone system in itself that in certain circumstances collaborates with the HSP90 system. Some more clarity in the definition of the terms ATP dependence, chaperone cycle, N-terminal domain (including the ATPase), C-terminal domain, and dimerization would be helpful.

HSP70

It would be important to discuss whether the compounds targeting ‘HSP70’ used and discussed in cancer therapy affect all 13 human HSP70 proteins, or only a subset. 

Specific comments:

Line 364: what does PES stand for?

Lines 372-374: These data suggest a way to overcome Hsp70-mediated proteostasis by increasing misfolded protein degradation. Both the HSP70 chaperone function and appropriate degradation of misfolded proteins are mechanisms promoting proteostasis. Does this mean that increasing degradation capacity balances higher chaperone protection of misfolded proteins (with cancer related mutations)?

Line 404: what is thermophoresis?

Lines 416-419: The data obtained with the aid of substrate-binding and refolding assays performed similarly to that in the search of AEAC (see above) allowed us to select the peptide showing those capacities and able to bind Hsp70; importantly the peptide ICyt2 elevated anti-cancer activity of doxorubicin in A431 epithelial carcinoma cells [106]. An example of a sentence, which is difficult to understand.

Author Response

We are grateful for the provided comments and propose the revision of our manuscript. According to the Editorial office comments we have revised our manuscript and have incorporated the changes.

This manuscript has the aim to review the literature on cancer treatment with a combination of molecular chaperone inhibitors and anti cancer drugs that address other targets used to counteract tumor growth and cancer cell survival. Most of the clinical, basal in vitro and in vivo studies discussed focus on heat shock factor 1, which is involved in stress-induced expression of heat shock and stress response proteins and the two chaperone systems HSP70 and HSP90. While the manuscript presents an ample record of the relevant literature, it is difficult to read as it lacks a clear structure and is superficial in describing mechanisms. The basic structure with the 2 headings and 4 subheadings is fine, but the text should focus more on what the abstract promises to deal with and be shortened combinatorial treatments involving chaperone-related drugs. Maybe a 3rd main heading summarizing/concluding at the end would also support readability and clarity. The many acronyms, short terms for cancer types and many terms just used once or twice are a challenge for the reader; some simplification, use of more descriptive terms, and avoiding using different terms for the same things would improve readability.

ANSWER: We have revised the main text of manuscript adding the new ‘3. Conclusion’ and ‘Abbreviations’sections.

There are many long and nested sentences, some in which the syntax appears odd. I am not a native speaker, but I think review by a native speaker and addition of commas would be helpful.

ANSWER: We have revised manuscript to correct the syntax.

Wording: The synonymous use of the terms’ combinational’, ‘combined’ and ’combinatorial’ is a bit confusing.

ANSWER: We revised this parts.

Overview

The first section is supposed to give an overview on combinatorial anti-cancer therapies. It starts with a general rationale for the advantages of combining drugs that target different cancer mechanisms. But it then follows tangents giving examples on which types of molecules are targeted, comments websites where one can look up the targets and effects of a given drug, and considerations and new techniques to deliver drugs more efficiently and increase stability. Before coming to the point, combining chaperoning mechanisms in combination with other targets, there is a lengthy and unconcise discussion of other targets. The first part of the last paragraph in section 1, without the list of examples, might instead be used to start this section. Then, continue shorter and more focused towards treatments with chaperone effectors plus another target.

The mechanistic rationale for targeting molecular chaperones and the difference of this target type from other anti-cancer therapies is not well defined. The same holds true for the introduction of misfolding protein aggregation, proteotoxic stress, and proteostasis.

Table 1 is in its current form not very helpful. As the focus is on chaperone plus other target drug, a table on such combinations would be better. Or have a table of combinatorial treatments with key instances where no chaperone-targeting therapy was used and a clearly separated second part or a separate table 2 with key examples for the HSF1, HSP90, HSP70, or HSP27 targeting reports. A simplified keyword for the targeted mechanism would also be helpful. The table(s) should also in the text be described as for example listing selected examples, and what the criteria for selection (different types, most successful, or else) were applied.

ANSWER: We have revised the text and added new Table 2 that describes key examples of combined therapies.

Specific comments to this section:

Lines 52-55: At the cellular level the drugs target mechanisms of protein modifications like phosphorylation, proteolysis or acetylation, while among the most desirable cellular targets are mitochondrial functions, membrane integrity, nuclear acid-protein complexes, cell transport mechanism, proteolytic machinery and others (Fig. 1). What is meant by desirable and how does this contrast to the mechanisms listed in the first part of the sentence?

ANSWER: We have corrected this sentence.

Lines 82-85: Similarly, metformin, an activator of AMP-dependent protein kinase metabolic stress machinery and commonly employed cisplatin notably reduced metastasis by affecting the epithelial-mesenchymal transition (EMT) in in vivo ovarian cancer models [13]. Activation of AMPK is just one of many reported effects of metformin; naming it here alone and without mentioning the target for cisplatin is insufficient.

ANSWER: We agree with a reviewer so we have deleted this paragraph from the manuscript.

HSF1

As stated in the abstract of the quoted reference (24), HSF1 is involved in regulation of heat shock and stress proteins, but also many functions and mechanisms that are not directly related to the accumulation of misfolding and aggregating proteins. Furthermore, not all HSPs are true heat shock regulated proteins. Especially for the HSP70 family, constitutively and stress regulated orthologs exist.

ANSWER: We agree with a reviewer and added this notion into the text.

The description of pro-tumoric effects of HSF1 and the rationale for the finding that triptolide is ‘specific’ is not well understandable. It ‘has been shown to efficiently inhibit the Hsp synthesis’. Does this mean synthesis of all/many proteins whose genes possess HSE’s? Why is this more specific than inhibiting HSF1 in other ways?

ANSWER: In the current review paper we focused on the Hsf1 inhibitors in relation to the HSPs production. Other effects of the Hsf1 inhibition we did not consider in the current paper.

HSP90

There are 4 human HSP90 orthologs, two cytosolic, one in the ER and one in mitochondria. Do the statements apply for all or only for the cytosolic ones that have the function to modify the conformations of transcription factors. HSP90 activity inhibitors likely affect them all. I would not consider HSP70 a co-chaperone of HSP90, rather a chaperone system in itself that in certain circumstances collaborates with the HSP90 system. Some more clarity in the definition of the terms ATP dependence, chaperone cycle, N-terminal domain (including the ATPase), C-terminal domain, and dimerization would be helpful.

ANSWER: We agree with a reviewer concerning the existence of four Hsp90 homologs that could influence the efficacy of the therapies. We have added this notion into the paper.

HSP70

It would be important to discuss whether the compounds targeting ‘HSP70’ used and discussed in cancer therapy affect all 13 human HSP70 proteins, or only a subset. 

 ASNWER: We have added this notion into the review.

Specific comments:

Line 364: what does PES stand for?

ANSWER: PES stands for 2-phenylethynesulfonamide

Lines 372-374: These data suggest a way to overcome Hsp70-mediated proteostasis by increasing misfolded protein degradation. Both the HSP70 chaperone function and appropriate degradation of misfolded proteins are mechanisms promoting proteostasis. Does this mean that increasing degradation capacity balances higher chaperone protection of misfolded proteins (with cancer related mutations)?

ANSWER: In our opinion Hsp70 inhibition from one hand increases protein degradation and from other hand also increases the presence of misfolded proteins that could influence the immunogenicity of the tumor cells.

Line 404: what is thermophoresis?

ANSWER: Microscale thermophoresis (MST) is a technology for the biophysical analysis of interactions between biomolecules. Microscale thermophoresis is based on the detection of a temperature-induced change in fluorescence of a target as a function of the concentration of a non-fluorescent ligand.

Lines 416-419: The data obtained with the aid of substrate-binding and refolding assays performed similarly to that in the search of AEAC (see above) allowed us to select the peptide showing those capacities and able to bind Hsp70; importantly the peptide ICyt2 elevated anti-cancer activity of doxorubicin in A431 epithelial carcinoma cells [106]. An example of a sentence, which is difficult to understand.

ANSWER: We have corrected this sentence.

Round 2

Reviewer 2 Report

The manuscript still needs editing for grammar and spelling.

Examples (but not limited to these): Line 527 is missing a "the"; Fig 2 legend has "Hsf!" typo.  These (and others) should both be caught by a grammar/spell checker. 

Author Response

Dear Editorial Office,

We are grateful for the provided comments and propose the revision of our manuscript. According to the reviewer comments we have revised our manuscript and have incorporated the changes.

Comments:

The manuscript still needs editing for grammar and spelling.

Examples (but not limited to these): Line 527 is missing a "the"; Fig 2 legend has "Hsf!" typo.  These (and others) should both be caught by a grammar/spell checker. 

ANSWER: With have corrected the grammar and syntax throughout the manuscript. Typo error at Line 527 and the Fig. 2 legend were also corrected.

Sincerely on behalf of all co-authors,

Dr. Maxim Shevtsov

M.D., Ph.D.

Reviewer 3 Report

All suggested edits appear to have been made. I am happy to see classification of inhibitors based on binding domains for each chaperone. This addition in the form of a figure(s) makes the information much more accessible to the reader. I can now recommend this manuscript for publication. 

Author Response

Dear Editorial Office,

We are grateful for the provided comments and propose the revision of our manuscript.

Comments:

All suggested edits appear to have been made. I am happy to see classification of inhibitors based on binding domains for each chaperone. This addition in the form of a figure(s) makes the information much more accessible to the reader. I can now recommend this manuscript for publication. 

ANSWER: On behalf of all authors I would like to thank the Reviewer for the provided comments and work with our manuscript.

Sincerely on behalf of all co-authors,

Dr. Maxim Shevtsov

M.D., Ph.D.

Reviewer 4 Report

The manuscript is significantly improved by the changes and additions. The second table, list of abbreviations, and clarification of sentences is positive. I still feel that the manuscript is difficult to read and may be difficult to access by a broader readership. But this may be difficult to avoid when collecting an overview over a lot of one sentence examples of different therapies and therapy combinations. Language is for the largest part fine, but some review of syntax and especially commas is advised. 

Author Response

Dear Editorial Office,

We are grateful for the provided comments and propose the revision of our manuscript. According to the Reviewer’s comments we have revised our manuscript and have incorporated the changes.

Comments: The manuscript is significantly improved by the changes and additions. The second table, list of abbreviations, and clarification of sentences is positive. I still feel that the manuscript is difficult to read and may be difficult to access by a broader readership. But this may be difficult to avoid when collecting an overview over a lot of one sentence examples of different therapies and therapy combinations. Language is for the largest part fine, but some review of syntax and especially commas is advised. 

ANSWER: On behalf of all authors I would like to thank the Reviewer for the provided comments and work with our manuscript. We have corrected the grammar and syntax throughout the manuscript.

Sincerely on behalf of all co-authors,

Dr. Maxim Shevtsov

M.D., Ph.D.